# Preclinical Assessment of Dactinomycin in *KMT2A*-Rearranged Infant Acute Lymphoblastic Leukemia

**DOI:** 10.3390/cancers17030527

**Published:** 2025-02-05

**Authors:** Sung K. Chiu, Emanuela Ferrari, Joyce Oommen, Sebastien Malinge, Laurence C. Cheung, Rishi S. Kotecha

**Affiliations:** 1Leukaemia Translational Research Laboratory, WA Kids Cancer Centre, The Kids Research Institute Australia, Perth, WA 6009, Australia; sung.chiu@telethonkids.org.au (S.K.C.); emanuela.ferrari@telethonkids.org.au (E.F.); joyce.oommen@telethonkids.org.au (J.O.); sebastien.malinge@telethonkids.org.au (S.M.); laurence.cheung@telethonkids.org.au (L.C.C.); 2Curtin Medical School, Curtin University, Perth, WA 6102, Australia; 3Medical School, University of Western Australia, Perth, WA 6009, Australia; 4Curtin Medical Research Institute, Curtin University, Perth, WA 6102, Australia; 5Department of Clinical Haematology, Oncology, Blood and Marrow Transplantation, Perth Children’s Hospital, Perth, WA 6009, Australia

**Keywords:** infant acute lymphoblastic leukemia, *KMT2A*-rearranged, drug screening, dactinomycin

## Abstract

Infant acute lymphoblastic leukemia (iALL) is an aggressive form of leukemia with a poor prognosis and few new treatment options. We used patient-derived iALL cell lines to perform an in vitro drug screen with existing anti-cancer agents with the aim to repurpose them for clinical use; we found dactinomycin to have significant activity at nanomolar concentrations. In vitro combination testing of dactinomycin with conventional chemotherapeutic agents used for the treatment of iALL identified a mild additive effect with cytarabine. However, in vivo treatment of dactinomycin in iALL patient-derived xenograft models led to a limited improvement in survival. These results indicate that dactinomycin is unlikely to provide clinical benefit for the treatment of iALL.

## 1. Introduction

Acute lymphoblastic leukemia (ALL) is the most common form of cancer in children, with an annual incidence of approximately 36 cases per million in the United States of America [1]. Over the past fifty years, the intensification of conventional chemotherapy has improved 5-year overall survival rates for children with ALL to over 90% [2]. However, infants with ALL (iALL), defined as patients diagnosed before their first birthday, continue to have significantly inferior survival outcomes [3,4]. Over 80% of iALL harbor a *KMT2A* rearrangement, which is an aggressive genetic driver mutation resulting in a 5-year event-free survival (EFS) of less than 45% [5,6,7,8,9,10,11,12,13,14]. While infants often respond to initial chemotherapy, there is a high risk of early relapse, with two-thirds of all relapses occurring within the first year of diagnosis. Outcomes from clinical trials for iALL have remained at a plateau over several decades, necessitating the identification of novel strategies for iALL. While one study has shown the benefit of early intensification for infants with *KMT2A*-rearranged ALL and reserving allogeneic hematopoietic stem cell transplant for those stratified as high risk [15], this has to be balanced with the knowledge that the use of intensive treatment modalities leave those who survive with lifelong complications [16]. The addition of blinatumomab to therapy holds significant promise, with a pilot study demonstrating a marked improvement in outcome compared with a historical control [17], leading to further investigation of this agent in the current suite of clinical trials for iALL.

Drug development remains a very challenging endeavor, with a median time of 13.5 years for a newly developed drug to enter clinical use at a median cost of USD 2.3 billion [18,19]. Therefore, there has been increasing interest in repurposing currently available drugs for novel indications. Given that these drugs are already in clinical use or undergoing testing in clinical trials, they have known safety and toxicity profiles, which can significantly reduce development time and cost. One of the best recent examples of drug repurposing is regarding the use of the Janus kinase (JAK) inhibitor baricitinib, which was originally indicated for the treatment of rheumatoid arthritis and was found to be efficacious for the treatment of severe COVID-19 [20]. Timely access to novel drugs also poses another significant issue for infants, given that confirmation of safety and efficacy in adults and older children is often required before consideration for investigation in younger children. Therefore, drug repurposing can potentially offer a much quicker path to identifying active agents for rare disorders such as iALL.

In this study, we performed in vitro testing of 62 anti-neoplastic agents that are currently in clinical use with the aim to identify agents that are cytotoxic to a unique panel of extensively characterized *KMT2A*-rearranged iALL cell lines. We identified dactinomycin as a lead candidate for further testing (also known as actinomycin D). Dactinomycin is derived from the bacteria *Streptomyces parvulus* and has been used for the treatment of cancers since the 1960s [21]. It is currently used to treat several commonly occurring childhood solid tumors. There are also case reports of dactinomycin demonstrating efficacy in relapsed/refractory ALL and acute myeloid leukemia (AML) [22,23]; however, it has never been assessed as a therapy for iALL. We performed detailed preclinical testing to determine the tolerability and efficacy of dactinomycin for its potential to treat iALL.

## 2. Materials and Methods

### 2.1. Cell Culture

A panel of 8 human cell lines was established as previously described from bone marrow or peripheral blood leukocytes of infants diagnosed with *KMT2A*-rearranged ALL at Perth Children’s Hospital, Western Australia [24,25]. All cell lines were developed following approval from the Child and Adolescent Health Service Human Research Ethics Committee (1769EP). Informed consent was obtained from all subjects and/or their legal guardians. Cells were maintained in RPMI 1640 (Thermo Fisher Scientific, Waltham, MA, USA) supplemented with 10% fetal calf serum (FCS; Cell Sera, Rutherford, NSW, Australia), L-glutamine (2 mM final concentration; Sigma-Aldrich, St. Louis, MO, USA), 2-mercaptoethanol (2 mM final concentration; Sigma-Aldrich, St. Louis, MO, USA), 1% non-essential amino acid mix (100×; MP Biomedicals, Santa Ana, CA, USA) and 1% 100 mM sodium pyruvate solution (MP Biomedicals, Santa Ana, CA, USA), hereafter referred to as complete media. Cell viability was assessed through live cell counting by the exclusion of 0.2% Trypan blue (Sigma-Aldrich, St. Louis, MO, USA). For all experiments, cells were cultured at 37 °C and 5% CO_2_. All cell lines were authenticated by short tandem repeat (STR) profiling by the Australian Genome Research Facility (AGRF, Melbourne, Australia) and repeatedly shown to be mycoplasma free using the MycoAlert™ PLUS Mycoplasma Detection Kit (Lonza, Basel, Switzerland).

### 2.2. In Vitro Drug and Sensitivity Testing

The anti-neoplastic drug panel tested was part of an FDA-approved drug library purchased from Compounds Australia (Griffith University, Australia). Test compounds were added to 384-well culture plates (Corning, NY, USA) at a concentration of 5 μM (in DMSO stock). Cells were seeded at pre-determined densities in 50 μL complete media using an automated Multidrop™ Combi Reagent Dispenser (Thermo Fisher Scientific, Waltham, MA, USA) [25]. Plates were incubated for 72 h at 37 °C and 5% CO_2_, after which Alamar Blue reagent comprising resazurin, methylene blue and potassium hexacyanoferrate (II and III) (Sigma-Aldrich, St. Louis, MO, USA) was added (10% *v*/*v*). Cell viability was determined by absorbance (570 nm and 600 nm) after 6 h using a Synergy™ Mx microplate reader (Biotek, Winooski, VT, USA). The percentage cell viability for each test compound was calculated relative to the cells treated with vehicle (100% viability).

Half-maximal inhibitory concentrations (IC_50_) for dactinomycin were determined in each iALL cell line by serial dilutions from 0.2 μM followed by incubation for 72 h and analyzed using the modified Alamar Blue assay. Calculations for IC_50_ were performed in Microsoft Excel.

For the synergy assays, we assessed each of the nine conventional chemotherapy agents used to treat iALL and dactinomycin, at 13 and 6 concentrations, respectively, using a series of 2-fold dilutions. The viability of each cell line was calculated for each drug concentration after an incubation of 72 h. Synergy was calculated by the Bliss Independence model using Synergy Finder [26,27,28].

### 2.3. Assessment of In Vivo Efficacy

Female NOD/SCID mice were purchased from the Animal Research Centre (Perth, Australia) and housed under pathogen-free conditions. Experimental protocols were approved by the Animal Ethics Committee, The Kids Research Institute Australia, Perth (AEC#373: Using dactinomycin to treat high risk infant leukemia; approved on 20 October 2020). All methods are reported in accordance with the ARRIVE guidelines for the reporting of animal experiments and were carried out in accordance with institutional guidelines and regulations. Mice aged between six to eight weeks were inoculated with cells from established iALL patient-derived xenograft (PDX) models, namely MLL-5 (which harbors the t(10;11) or KMT2A/MLLT10 translocation) at 1 × 10^6^ cells per mouse or MLL-7 (which harbors the t(4;11) or KMT2A/AFF1 translocation) at 2 × 10^6^ cells per mouse [29]. Dactinomycin was reconstituted in 5% DMSO, 40% PEG300, 5% Tween-80 and purified water. Dactinomycin (or vehicle control) treatment started when the percentage of human CD19 and CD45 co-expressing cells reached > 1% in the peripheral blood, as measured by weekly tail vein bleeding. Whole blood was collected via tail puncture from the mice, red cells were lysed for 30 min at 4 °C (1xRBC Lysis Buffer, Invitrogen, Carlsbad, CA, USA, Cat#00-4333-57), washed twice with cold phosphate buffered saline (PBS) (Gibco, Thermo Fisher Scientific, Waltham, MA, USA, Cat#10010023-1) and stained with 2 µL of anti-human CD45-APC (Becton Dickinson, Franklin Lakes, NJ, USA, Cat#555485) and 2 µL of anti-human CD19-PE (Becton Dickinson, Franklin Lakes, NJ, USA, Cat#560728) in a final volume of 100 µL of PBS for half an hour in the dark at 4 °C. Cells were then washed once with cold PBS, resuspended in 100 µL of 2% FCS in PBS and analyzed by flow cytometer. Half a million events were acquired and gated against the unstained control.

The maximum tolerated dose (MTD) was determined in the MLL-5 PDX model using concentrations of dactinomycin ranging from 18 μg/kg to 300 μg/kg. For EFS studies, MLL-5 and MLL-7 mice were divided into groups and treated over 4 weeks with dactinomycin at the MTD and at the dose level below the MTD or vehicle control. Leukemia progression was monitored by measuring the percentage of CD45^+^CD19^+^ cells in the blood weekly, and EFS was calculated from the day of treatment initiation until mice reached a pre-determined humane endpoint with evidence of leukemia-related morbidity. Once this pre-determined endpoint was reached, mice were euthanized by isoflurane followed by cervical dislocation.

### 2.4. Statistical Analyses

Figures and graphs were produced using GraphPad Prism version 10.4.0 (Boston, MA, USA), with data analyses performed using a two-tailed unpaired Student’s *t*-test. Survival studies were analyzed using the log-rank test. A *p*-value of < 0.05 was considered statistically significant.

## 3. Results

### 3.1. Anti-Neoplastic Drug Screen Identifies Activity of Dactinomycin in Infant ALL

To identify drugs that can potentially be used for the treatment of iALL, we initially performed a cancer drug screen comprising 62 agents at a single concentration of 5 μM using 8 *KMT2A*-rearranged iALL cell lines generated from, and shown to be representative of, the originating primary patient samples [24,25]. The results of the drug screen are shown in Figure 1. A number of agents that are currently in use for the treatment of iALL were identified as being active against the iALL cell lines, including dexamethasone, anthracyclines such as daunorubicin and anti-tubulin agents such as vincristine. The drug screen identified dactinomycin as one of the top-ranked candidates and, given that it currently forms part of the standard treatment for childhood cancers such as Wilms tumor, Ewing sarcoma and rhabdomyosarcoma, it was selected for further testing for iALL.

### 3.2. In Vitro Assessment of Dactinomycin in Infant ALL

Consistent with results from our initial drug screen, dactinomycin was highly active in vitro and demonstrated low IC_50_ concentrations in the nanomolar range (PER-490: 0.8 nM, PER-494: 1.2 nM, PER-784: 1.6 nM, PER-785: 0.48 nM, PER-826: 1.1 nM and PER-910: 0.47 nM) (Figure 2). Combination in vitro drug testing with the nine conventional chemotherapeutic agents used to treat iALL demonstrated a mild additive effect with cytarabine, whereas there was heterogeneity in the combinatorial response between cell lines when tested with the other chemotherapy agents (Table 1).

### 3.3. Dactinomycin Has a Minimal In Vivo Survival Benefit

Mice transplanted with MLL-5 cells were initially treated with doses of dactinomycin from 75 μg/kg to 300 μg/kg to determine the maximum tolerated dose (MTD); they exhibited significant toxicity, with reduced movement and significant weight loss, and over 50% of mice required euthanasia after the first dose. We therefore tested reduced dosages of dactinomycin and the MTD was determined to be 36 μg/kg as this did not result in the death of mice within 48 h. This dose was used as the MTD for subsequent survival studies.

In order to evaluate the efficacy of dactinomycin, mice transplanted with MLL-5 were treated with either 18 μg/kg (once, twice or four times a week) or 36 μg/kg (once or twice weekly) of dactinomycin over 4 weeks. This led to a modest improvement in survival; mice treated with 18 μg once, twice or four times a week showed the best results, with a median survival of 50 days vs. 45 days for vehicle-treated mice (*p* = 0.003). Mice treated with 36 μg/kg once weekly showed a median survival of 49 days (*p* = 0.003) and those treated with 36 μg twice weekly showed a survival benefit of only 1 day compared with vehicle controls (46 days; *p* = 0.003) (Figure 3A). To further consolidate this finding, we conducted the same experiment using the MLL-7 PDX model. Similarly, our results showed a statistically significant improvement in the median survival for mice treated with 18 μg/kg once weekly (49 vs. 43 days; *p* = 0.002) or twice weekly (59 vs. 43 days; *p* = 0.002) compared with vehicle-treated mice. However, there was no survival advantage using the 18 μg/kg four times a week dosing (40 vs. 43 days; *p* = 0.001) and no significant difference in survival outcome for mice treated with 36 μg/kg once weekly (41 vs. 43 days; *p* = 0.05) or 36 μg/kg twice weekly (46 vs. 43 days; *p* = 0.14) compared with vehicle-treated mice (Figure 3B).

Results for the 18 μg/kg four times a week arm were discrepant between the two PDX models tested. For most treatment arms, assessment at the time of euthanasia demonstrated high leukemia burden in the bone marrow, indicating that mice succumbed to leukemia. However, in MLL-7 the bone marrow leukemia burden at endpoint was much lower in the 18 μg/kg four times a week group, suggesting that toxicity may have contributed to early mortality (Appendix A).

Overall, given that the benefit identified for dactinomycin appeared limited, further evaluation of this agent was not pursued.

## 4. Discussion

Treatment of infants with *KMT2A*-rearranged ALL continues to remain a significant challenge. The distinct biology leads to aggressive disease and high rates of relapse. The number of novel therapeutic agents that can be assessed in clinical trials is limited by the rarity of the patient population, highlighting the need for robust preclinical studies to identify new treatment approaches. While the translational focus should be the identification of agents that have a high potential to be clinically effective, it is equally essential to report studies where there is a marginal or negative effect. This is to ensure that these agents are not prioritized for clinical evaluation and also to avoid the repetition of experimental work that is known to be futile.

Using our established drug-screening pipeline, we evaluated a library of FDA-approved anti-neoplastic agents that are currently in clinical use. A number of agents exhibited cytotoxicity in our iALL patient-derived cell lines, with dactinomycin selected for further testing for several reasons. Dactinomycin has a long history of use as an anti-neoplastic agent; it was first developed in the 1960s and is known to bind DNA, leading to the inhibition of RNA-polymerase I and RNA transcription [21,30]. Dactinomycin is a low-cost agent that is readily translatable to infants with ALL, given that it is currently in clinical use as part of combinatorial multi-agent therapy for the treatment of certain pediatric solid tumors, such as Wilms tumor, Ewing sarcoma and rhabdomyosarcoma. Notably, dactinomycin has been used extensively in the infant population and has a well-established dosing, schedule and toxicity profile, making it an attractive candidate for repurposing in infant ALL. Furthermore, in the setting of leukemia, a recent single-agent study of dactinomycin administered at a dose of 15 μg/kg for 5 consecutive days every 2 to 4 weeks in adults with relapsed/refractory *NPM1*-mutated AML demonstrated significant clinical activity, with four out of nine patients demonstrating a complete response [23]. Support for further investigation in the setting of iALL was evidenced through the independent identification of dactinomycin as one of the top cytotoxic agents in a high throughput ex vivo drug screen of iALL PDX samples in a recent publication [31].

Dactinomycin has undergone in vivo investigation for a wide variety of cancers, including leukemias, brain tumors and gastrointestinal tumors, with variable doses ranging from 25 μg/kg daily to 600 μg/kg weekly, with the use of intraperitoneal, intravenous and direct intra-tumoral administration [32,33,34,35]. Our iALL PDX models showed a much narrower therapeutic index of dactinomycin compared with other published studies, which may reflect intrinsic differences in preclinical cancer models, including the strain of immunodeficient mice used for transplantation. In our in vivo study, we identified the MTD of dactinomycin as being 36 μg/kg. This was based on the acute effects in mice treated with higher doses resulting in significant toxicity necessitating euthanasia within 24 h. However, given that the mice treated with doses of 18 μg/kg derived greater survival benefit than the mice treated with doses of 36 μg/kg, it is likely that some unrecognized longer term toxicities may also manifest with dosing at the MTD and with higher cumulative dosing strategies.

A significant survival advantage was seen in mice treated with 18 μg/kg dactinomycin compared with vehicle-treated mice. The standard dactinomycin dose for cancer treatment in humans varies according to the type of cancer and population being treated. Notably, for infants treated according to the Children’s Oncology Group protocols, a 23 μg/kg/dose has been used for the treatment of Wilms tumor, whereas, in rhabdomyosarcoma, weight-based dosing is used for patients less than 14 kg, ranging from 90 to 600 μg depending on body weight as part of multi-modal chemotherapy. Thus, in our study, we used a dosing strategy comparable to that which is currently used to treat infants with cancer. However, the effects of the 18 μg/kg/dose were modest at best, with an improved survival of between 5 and 16 days. This may reflect the inability of dactinomycin to inflict a dramatic response as a single agent due to the aggressive nature of *KMT2A*-rearranged ALL. It is possible that the efficacy of dactinomycin could be improved with combination therapy; however, our in vitro synergy data did not indicate a clear candidate that was worth pursuing for in vivo combination testing.

In interpreting the results of our study, it is important to note that we only used a single concentration of 5 μM for all of the anti-cancer compounds in our initial drug screen. The intent of this initial assessment was to provide the rationale for identifying drugs that can be taken forward for further investigation; this does not preclude the possibility that the drugs tested may be effective at a higher concentration. In addition, the compounds tested all have different modes of action and not all will necessarily have a direct cytotoxic effect; more detailed combination testing of our drug panel may yet uncover synergistic combinations between agents with limited single-agent activity in our initial drug screen.

Another consideration is that we included both drug toxicity and leukemia-induced morbidity as events for our in vivo assessments. During determination of the MTD, the attribution to drug toxicity was clear with dactinomycin doses ≥ 75 μg/kg, based on severe clinical symptomatology necessitating euthanasia. This differentiation was less evident when performing the EFS studies. For most treatment arms, given the high percentage of leukemia cells in the bone marrow at the time of death, we inferred that the primary cause of death was related to leukemia. However, the bone marrow burden at endpoint was much lower in MLL-7 mice treated with 18 μg/kg of dactinomycin four times a week, indicating that drug toxicity may also have played a role in this group. Serial assessment of blood parameters and detailed necropsy at the time of euthanasia may have assisted with further delineation.

## 5. Conclusions

In conclusion, the preclinical assessment of dactinomycin for iALL demonstrated marked in vitro cytotoxicity, which led to marginal survival benefit in vivo. Dactinomycin does not therefore represent a priority candidate for integrating into therapy for infants with ALL and alternative novel agents should be considered for clinical use.

## Figures and Tables

**Figure 1 cancers-17-00527-f001:**
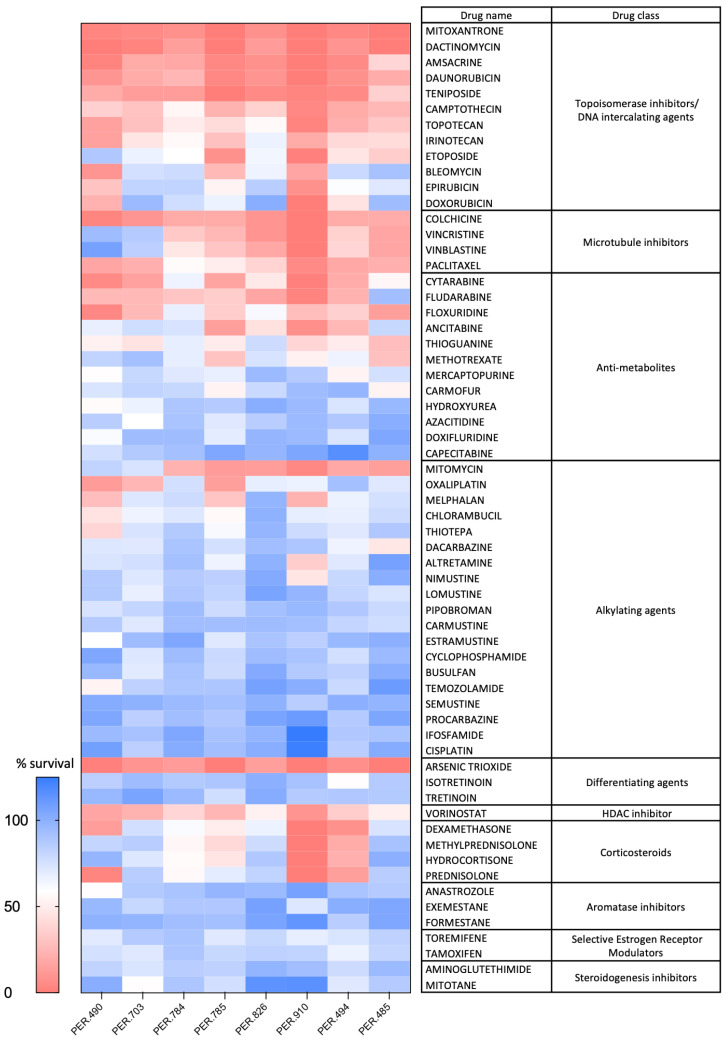
Heatmap showing survival of infant acute lymphoblastic leukemia cell lines incubated for 72 h with an FDA-approved anti-neoplastic drug screen at a concentration of 5 µM. The percentage survival represents cell viability relative to cells treated with vehicle control. Each square represents a single experiment.

**Figure 2 cancers-17-00527-f002:**
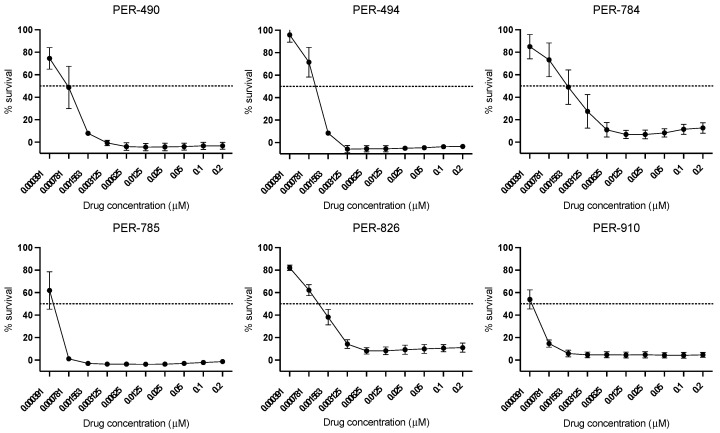
Dose–response analyses of dactinomycin in PER cell lines derived from infants with *KMT2A*-rearranged ALL. Half-maximal inhibitory concentrations (IC_50_) were determined in each iALL cell line by serial dilutions from 0.2 μM followed by incubation for 72 h and analyzed using the modified Alamar Blue assay. Data points are represented as the mean ± standard deviation of *n* = 9 replicates.

**Figure 3 cancers-17-00527-f003:**
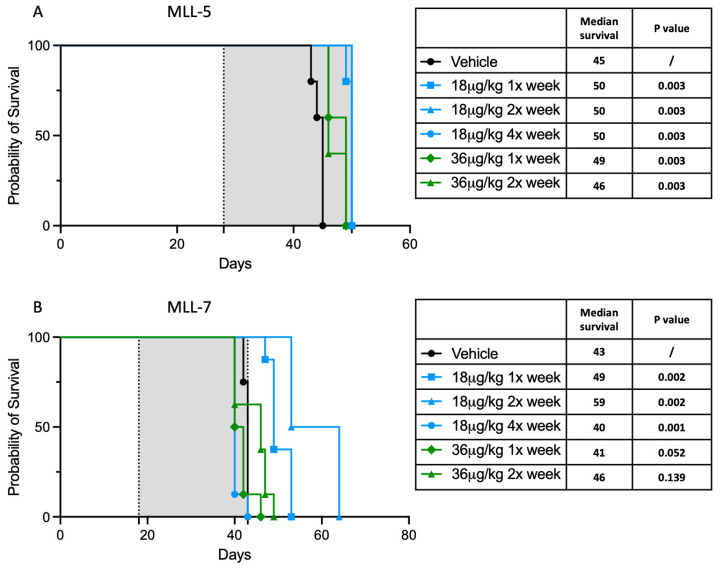
Treatment of infant *KMT2A*-rearranged patient-derived xenografts with dactinomycin. Kaplan–Meier survival curves of mice injected with (**A**) MLL-5 or (**B**) MLL-7 leukemia cells that were treated with dactinomycin for 4 weeks at various doses and frequencies. *p*-values derived from log-rank (Mantel–Cox) test compared with vehicle. Grey shaded areas indicate treatment duration. Between six and eight mice were used for each arm of the experiment.

**Table 1 cancers-17-00527-t001:** Total in vitro synergy scores between dactinomycin in combination with conventional chemotherapy agents. Each experiment was performed in triplicate with mean results reported.

Cell Lines	Vincristine	Daunorubicin	Dexamethasone	Cytarabine	Methotrexate	Mercaptopurine	L-Asparaginase	Thioguanine	4-Hydroxy-Cyclophosphamide
PER-490	−1.94	−2.77	−0.08	1.59	−1.43	−2.13	−1.17	−2.91	−0.49
PER-494	0.34	−3.29	0.71	3.42	−6.01	−6.68	−0.86	−7.21	−3
PER-785	0.83	0.92	−1.19	0.51	0.07	0.75	1.61	1.08	0.77
PER-826	−4.68	−4.48	−2.5	−0.8	−6.27	−3.17	−1.76	−1.9	−2.87
PER-910	0.68	−0.44	0.41	1.77	0.23	−1.85	−3.46	−1.55	0.51
PER-784	−6.65	−4.47	−6.04	0.62	−9.9	−2.7	−6.74	−5.61	−3.67
Colour key:									
<−10	Antagonistic								
−10 to 10	Additive								
>10	Synergistic								

## Data Availability

The datasets used and/or analyzed during the current study are available from the corresponding author on reasonable request.

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
