# Peer review of "Preclinical Assessment of Dactinomycin in KMT2A-Rearranged Infant Acute Lymphoblastic Leukemia"

_cancers, 2025, doi:10.3390/cancers17030527_

Round 1

Reviewer 1 Report

Comments and Suggestions for Authors

In this study, the authors focus on infant acute lymphoblastic leukemia (iALL), a subgroup characterized by specific genetic abnormalities and inferior outcome, emphasizing the need for novel treatment strategies. Therefore, the authors employ a compound screen testing 62 drugs established for cancer treatment at a single concentration in order to identify active drugs against iALL to be potentially re-purposed for iALL therapy. Fore the screening n=8 cell lines derived from infant ALL patients established in the authors institution are used. As one drug showing high anti-iALL activity the chemotherapeutic dactinomycin was found and further evaluated addressing synergies with other chemotherapeutics and pre-clinical activities using patient-derived xenograft mouse models.

As a result, and despite to identification in the screen, dactinomycin did not show a synergism with one of the drugs investigated and no clear pre-clinical activity against iALL in vivo.

1. The authors used identical concentrations of 5 microM of the different compounds and drugs. How do these concentrations relate to concentrations achieved in clinical application in patients? Different classes of substances and drugs with different modes of action are included into the screening - how do the authors see comparabilities of the in vitro effects?

2. Figure 1: a legend explaining the meaning of the different color-coding (% reduction of metabolic activity?) is missing.

3. Figure 3/survival analyses: all events correspond to morbidity of ALL or also toxicity-associated morbidities included? How was this assessed?

Author Response

Comment 1: The authors used identical concentrations of 5 microM of the different compounds and drugs. How do these concentrations relate to concentrations achieved in clinical application in patients? Different classes of substances and drugs with different modes of action are included into the screening - how do the authors see comparabilities of the in vitro effects?

Response 1: Thank you for this comment. We agree with the reviewer that the drug screen performed at a concentration of 5 micromolar is not intended to relate to concentrations achieved in patients. The intent of this initial assessment was to provide rationale for identifying a drug that we could carry forward for further investigation.

From these results, we selected dactinomycin for further investigation and were then able to perform comprehensive dose response analyses of this drug at different concentrations. Notably, we were able to investigate similar in vivo doses of dactinomycin to that which has already been investigated in the clinical setting for different indications, as highlighted in the discussion.

We also acknowledge that the drug screen is not intended to compare between agents as different drugs will have different modes of action and not all will necessarily have a direct cytotoxic effect. As such, we have not made any recommendations to exclude investigation of drugs which did not demonstrate cytotoxicity from our initial screen. Open access publication of the screen therefore provides the opportunity for other researchers to investigate some of the other drugs which may be of interest.   

Comment 2: Figure 1: a legend explaining the meaning of the different color-coding (% reduction of metabolic activity?) is missing. 

Response 2: Thank you for highlighting this omission. Figure 1 has been corrected to include the colour key.

Comment 3: Figure 3/survival analyses: all events correspond to morbidity of ALL or also toxicity-associated morbidities included? How was this assessed?

Response 3: Thank you for your comment. Survival in Figure 3 demonstrates death from any cause. Post-euthanasia we analysed the bone marrow and spleen to determine the percentage of leukaemia cells present by flow cytometry. Mice which demonstrated clinical signs of toxicity with drug treatment, such as reduced movement and significant weight loss, were not able to complete treatment and had to be euthanised early. They exhibited low levels of leukaemia cells in the bone marrow and spleen, indicating that they died from toxicity rather than disease. Mice without clinical symptoms were able to complete drug treatment and had a significant level of leukaemia (i.e. > 40%) at endpoint, indicating that they died from ALL.

Reviewer 2 Report

Comments and Suggestions for Authors

Authors tested the effects of several chemotherapeutic drugs on infant ALL cell lines with KMT2A mutation. Dactinomycin showed the best response in all cell lines. They determined the IC50 concentration of this drug on the cell lines in vitro, then determined the effective dose in vivo mouse model. They found it   is not so effective in vivo than in vitro.

Comments and questions:

1. Authors mentioned in the methods section (page 4 line 145) CD45+Cd19+ cell number was monitored in blood. How? Please describe the whole method!

2. Statistical section is missing!

3. In Figure 1. the color precedence is missing. Why mitoxantrone was not chosen? It was also similarly effective as dactinomycin.

4. Figure 2. what method was used to determined IC50? Please write the name of the method into the legend.

5. In table 5. there are some wrong numbers in the color key part.

6. Authors wrote in results section (page 6 line 186) 36ug/kg was determined, but how? They also mentioned this value in line 183 but is not clear from where does this value come?

7. How can the authors explain the difference between the in vivo and in vitro effect of Dactinomycin?

Author Response

Comment 1: Authors mentioned in the methods section (page 4 line 145) CD45+Cd19+ cell number was monitored in blood. How? Please describe the whole method!

Response 1: Thank you for your comment. We agreed with this suggestion and we have added a detailed description of flow cytometry analysis of leukaemia in tranpslanted mice in our methods section (page 3 line 141) as follows:

Whole blood was collected via tail puncture from the mice, red cells were lysed for 30 minutes at 4ËšC (1xRBC Lysis buffer, Invitrogen, California, USA, cat# 00-4333-57), washed twice with cold phosphate buffered saline (PBS) (Gibco, Thermo Fisher Scientific, Massachusetts, USA, Cat# 10010023-1) and stained with 2µl of anti-human CD45-APC (Becton Dickinson, New Jersey, USA, Cat# 555485) and 2µl of anti-human CD19-PE (Becton Dickinson, New Jersey, USA, Cat# 560728) in a final volume of 100µl of PBS for half an hour in the dark at 4ËšC. Cells were then washed once with cold PBS, resuspended in 100µl of 2% FCS in PBS and analyzed by flow cytometer. Half a million of events were acquired and gated against the unstained control.

Comment 2. Statistical section is missing!

Response 2: 

Response 2: Thank you for identifying this omission. We have added the statistical analyses section to our methods (page 4 line 160) as follows:

Statistical analyses and graphics were performed using GraphPad Prism version 10.4.0 (Boston, MA, USA). Data were analyzed using the two-tailed unpaired Student’s t-test. Survival studies were analyzed using log-rank test. A p value <0.05 was considered statistically significant.

Comment 3: In Figure 1. the color precedence is missing. Why mitoxantrone was not chosen? It was also similarly effective as dactinomycin.

Response 3: Thank you for  identifying this omission. We have corrected Figure 1 to include the colour key (page 5). We have also included the following to explain our colour scale (page 5 line 179):

The percentage survival represents cell viability relative to cells treated with vehicle control.

Mitoxantrone was not chosen for preclinical testing in our study as it is already used in the clinical setting as part of the ADE/MAE blocks of chemotherapy within the Interfant protocol and thus preclinical investigation would not provide further clinical benefit.

Comment  4: Figure 2. what method was used to determined IC50? Please write the name of the method into the legend.

Response 4: Thank you for your comment. A description of IC50 determination has now been added to the legend of Figure 2 (page 6 line 192) as follows:

Half maximum inhibitory concentrations (IC50) were determined in each iALL cell line by serial dilutions from 0.2μM followed by incubation for 72 hours and analyzed using the modified Alamar Blue assay.

Comment 5: In table 1. there are some wrong numbers in the color key part.

Response 5: Thank you for highlighting this discrepancy. We have corrected the colour within each cells of table 1 to more accurately reflect the colour key.

Comment 6: Authors wrote in results section (page 6 line 186) 36ug/kg was determined, but how? They also mentioned this value in line 183 but is not clear from where does this value come?

Response 6: Thank you for your comment. We identified the dose of 36ug/kg to be the maximum tolerated dose as this was the highest dose that did not result in death of our mice within 48 hours. The results section (page 6 line 203) has been modified to explain this more clearly as follows:

Mice transplanted with MLL-5 cells were initially treated with doses of dactinomycin from 75ug/kg to 300ug/kg to determine the maximum tolerated dose (MTD); they exhibited significant toxicity, with reduced movement and significant weight loss and over 50% of mice required euthanasia after the first dose. We therefore tested reduced dosages of dactinomycin and the MTD was determined to be 36ug/kg as this did not result in death of mice within 48 hours. This dose was used as the MTD for subsequent survival studies.

Comment 7: How can the authors explain the difference between the in vivo and in vitro effect of Dactinomycin?

Response 7: Thank you for your comment. The reasons for the discrepancy between the in vitro and in vivo effects of dactinomycin in infant ALL preclinical models are likely multifactorial. The dose-limiting toxicities that we saw in our mice at daily doses higher than 36ug/kg provide a limitation on drug levels achieved in vivo compared those which could be obtained in vitro. In addition, the discrepant results may reflect the inherent differences between in vitro and in vivo systems in cancer biology; in particular the bone marrow microenvironment may interact with leukaemia cells and promote quiescence and treatment resistance, which is a limitation of in vitro drug screening platforms. Further information on this topic can be found in our published review article (Panting RG, Kotecha RS, Cheung LC. The critical role of the bone marrow stromal microenvironment for the development of drug screening platforms in leukemia. Experimental Hematology 2024).

Reviewer 3 Report

Comments and Suggestions for Authors

The Authors report their preclinical experience with drug repurposing od d-actinomycin in KMT2A rearranged infant acute lymphoblastic leukemia. 

It is certainly of utmost importance focusing on drug repurposing in such a serious disease with limited possibility of cure different from trends showed by lymphoblastic leukemia of childhood. However I believe that the lack of efficacy of the tested drug, namely d-actinomycin,  limits the impact of the reported results. I agree that it is important to share also not significant results, but unavoidably it is of lower resonance. Moreover in the introduction, at the end of the first paragraph (line 57) when the authors state that "Outcomes from clinical trials for iALL have remained at a plateau over several decades[6], necessitating the identification of novel therapies for iALL."  I believe that it must be mentioned the excellent results of blinatumomab reported in this subset of leukemia:  "Two-year disease-free survival was 81.6%"  see  "van der Sluis IM et al. Blinatumomab Added to Chemotherapy in Infant Lymphoblastic Leukemia. N Engl J Med. 2023 Apr 27;388(17):1572-1581".

The paper is well written and the results are clearly presented, the discussion is short and correctly designed, but globally considered the paper does not gain such a high scientific value.

Author Response

Comment 1: I believe that it must be mentioned the excellent results of blinatumomab reported in this subset of leukemia:  "Two-year disease-free survival was 81.6%"  see  "van der Sluis IM et al. Blinatumomab Added to Chemotherapy in Infant Lymphoblastic Leukemia. N Engl J Med. 2023 Apr 27;388(17):1572-1581".

Thank you for your comment. We agree that blinatumomab has high potential to transform the care of patients with infant ALL following the results of this phase 2 study. We have restructured our introduction section to include this study (page 2 line 57) as follows:

While one study has shown benefit of early intensification for infants with KMT2A-rearranged ALL and reserving allogeneic hematopoietic stem cell transplant for those stratified as high risk [16], this has to be balanced with the knowledge that the use of intensive treatment modalities leave those who survive with lifelong complications [5]. The addition of blinatumomab to therapy holds significant promise with a pilot study demonstrating marked improvement in outcome compared to historical control [17], leading to further investigation of this agent in the current suite of clinical trials for iALL.

Reviewer 4 Report

Comments and Suggestions for Authors

Infant KMT2A-rearranged ALL remains a disease with poor prognosis in which no target therapies are available at present. In this work the authors investigated the “in vitro” potential therapeutic role of 62 anti-cancer drugs in eight infant ALL cell lines with KMT2A rearrangement and identified Dactinomycin as a promising drug with high activity at nanomolar concentrations. However, its use in xenograft models failed to demonstrate a significative improvement in survival, and lead to conclusion that Dactinomycin was unlikely to provide clinical benefit. 

The paper is interesting, experiments clear and well described, as well figures, references updated. 

I have just few observation/curiosity:

·      In “in vivo” experiments Dactinomycin was used as single agent, but ALL therapy results depends from the combination and sequential use of many agents: I am not completely sure that the modest survival improvement justify the conclusions. Did they tried combination experiments, i.e. with cytarabine? 

·      Did they observed differences according to KMT2A rearrangement? 

·      Why did they used different inoculation doses in in vitro experiments? Could this fact have affected results?

·      Could the lack of survival advantage in experiments with higher doses depend to off target toxicity? Did they evaluated organ, epithelial toxicity after mice euthanasia?

·      Did the authors have idea of the toxicity on normal hematopoietic stem cells?

·      Finally, from heat map in figure 1, also arsenic trioxide seemed to be active. Did they consider its potential use?

Author Response

Comment 1:  In “in vivo” experiments Dactinomycin was used as single agent, but ALL therapy results depends from the combination and sequential use of many agents: I am not completely sure that the modest survival improvement justify the conclusions. Did they tried combination experiments, i.e. with cytarabine? 

Response 1: Thank you for your comment. We agree that we observed mild in vitro additive effects between dactinomycin and cytarabine in our iALL cell lines with synergy scores ranging between -0.8 to 3.42. However, given the marginal in vivo efficacy of dactinomycin as a single agent we did not pursue combination studies with cytarabine in vivo as any marginal combinatorial gains identified would not be considered relevant for translation into the clinical setting. Infant ALL is a rare disease and so limits the number of clinical trials that can be conducted at any one time. Thus novel agents which show a high degree of efficacy from preclinical studies are prioritised for clinical investigation compared to those which only demonstrate marginal gains.

Comment 2: Did they observed differences according to KMT2A rearrangement? 

Response 2: Thank you for your comment. We did not observe any significant differences in response according to the KMT2A rearrangement.  A wide variety of different KMT2A-rearragements are represented by the cell lines in our in vitro panel (see Cheung LC, de Kraa R et al. Front Oncol 2021 for detailed cytogenetic characterisation of the cell lines) and the two models used for our in vivo studies (MLL-5 (t(10;11)) and MLL-7 (t(4;11)).

Comment 3: Why did they used different inoculation doses in in vitro experiments? Could this fact have affected results?

Response 3: Thank you for your comment. MLL-5 and MLL-7 are patient derived xenograft (PDX) models that were developed by Professor Richard Lock for use in the National Cancer Institute (NCI) supported Pediatric Preclinical In Vivo Testing Program (https://ctep.cancer.gov/MajorInitiatives/Pediatric_PIVOT_Program.htm). The cell doses used are required to ensure appropriate engraftment of the PDX model. The different cell doses for innoculation of MLL-5 and MLL-7 result in a median survival of untreated mice of around 40 days. As we innoculated the same dose of leukaemia cells within each arm of study according to the model used, we do not anticipate this to have affected the results.

Comment 4: Could the lack of survival advantage in experiments with higher doses depend to off target toxicity? Did they evaluated organ, epithelial toxicity after mice euthanasia?

Response 4: Thank you for your comment. At endpoint, we analysed the bone marrow and spleen of all euthanised mice by flow cytometry and determined them to have a significant level of leukaemia (i.e. >40%), indicating that they died from ALL. We agree with the potential of identifying additional toxicities if other organs were examined at necropsy after euthanasia, however this unfortunately was not performed at the time.

Comment 5: Did the authors have idea of the toxicity on normal hematopoietic stem cells?

Response 5: Thank you for your comment. We did not specifically examine the direct effect of dactinomycin on haematopoietic stem cells. However, we did measure haemoglobin, white cell and platelet counts on blood samples collected from mice treated with dactinomycin and there were no significant differences compared to vehicle treated mice.

Comment 6: Finally, from heat map in figure 1, also arsenic trioxide seemed to be active. Did they consider its potential use?

Response 6: Thank you for your comment. We agree that arsenic trioxide is another promising agent identified from our drug screen. We did consider it for further testing but were unable to proceed due to restrictions from our Institute which ascribed a high risk of toxicity with handling arsenic trioxide for preclinical testing in our laboratory environment. Our initial drug screen was purchased from Compounds Australia with all compounds inserted into 384-well plates and so no handling of arsenic was required from our end for the initial drug screen.  

Round 2

Reviewer 1 Report

Comments and Suggestions for Authors

#1 This point has been addressed sufficiently, but the revised ms has not been updated accordingly. However, the explanations given need to be included into the revised ms to clarify these points for all readers: please add the description of the intention to carry out the drug screen with one concentration only to the ms and include a critical discussion of the limitations of this approach. 

#2 The color key added to figure 1 is blank in the version provided for download, please correct. 

#3 As outlined in the reviewer’s response, both toxicity- and leukemia morbidity-induced deaths are included as events (‘Survival in Figure 3 demonstrates death from any cause’). Or are the toxicity deaths censored? If not, this means that (no) differences of survival are determined by drug ineffectivity and/or toxicity. This is an important piece of information and needs to be included: Given the relevance of the cause of death for the conclusion of the work (no candidate for further testing), mice died/sacrificed either due to toxicity or due to leukmia should be indicated and the impact of the respective reason analyzed. It could be, that despite good activity dactinomycin showed too high toxicity in this model and dosing, thereby resulting in no survival benefit. On the other hand, no anti leukemia activity while showing good treatment tolerability would lead to the same results. Or (probably) both mechanisms contribute? This information needs to be included and discussed. 

Author Response

Comment 1: This point has been addressed sufficiently, but the revised ms has not been updated accordingly. However, the explanations given need to be included into the revised ms to clarify these points for all readers: please add the description of the intention to carry out the drug screen with one concentration only to the ms and include a critical discussion of the limitations of this approach. 

Response 1: 

Thank you for your comments. We have added the following to our manuscript:

  1. In the results section (page 4, line 165), we have added ‘at a single concentration of 5mM’ to make it clear to the reader that only one concentration was tested in our initial drug screen.
  2. We have also added the following to the discussion section (page 8, line 282):

‘In interpreting the results of our study, it is important to note that we only used a single concentration of 5mM for all of the anti-cancer compounds in our initial drug screen. The intent of this initial assessment was to provide rationale for identifying drugs that can be taken forward for further investigation; this does not preclude the possibility that the drugs tested may be effective at a higher concentration. In addition, the compounds tested all have different modes of action and not all will necessarily have a direct cytotoxic effect; more detailed combination testing of our drug panel may yet uncover synergistic combinations between agents with limited single-agent activity in our initial drug screen.’  

Comment 2:  The color key added to figure 1 is blank in the version provided for download, please correct. 

Response 2: Thank you for noticing this omission. The format of the figure has been corrected to show the colour scale in PDF version.

Comment 3: As outlined in the reviewer’s response, both toxicity- and leukemia morbidity-induced deaths are included as events (‘Survival in Figure 3 demonstrates death from any cause’). Or are the toxicity deaths censored? If not, this means that (no) differences of survival are determined by drug ineffectivity and/or toxicity. This is an important piece of information and needs to be included: Given the relevance of the cause of death for the conclusion of the work (no candidate for further testing), mice died/sacrificed either due to toxicity or due to leukmia should be indicated and the impact of the respective reason analyzed. It could be, that despite good activity dactinomycin showed too high toxicity in this model and dosing, thereby resulting in no survival benefit. On the other hand, no anti leukemia activity while showing good treatment tolerability would lead to the same results. Or (probably) both mechanisms contribute? This information needs to be included and discussed. 

Response 3: Thank you for your comment. We agree that both toxicity- and leukemia-induced morbidity were included as events in our in vivotesting of dactinomycin. With regard to Figure 3, we performed end organ assessments of euthanised mice and showed that the leukaemia burden in the bone marrow was greater than 50%, inferring that the most likely cause of death was disease related.

 To address these comments in our manuscript, we have made the following changes:

  1. We have added Supplemental Figure 1 to show the percentage of leukaemia cells in the blood, bone marrow and spleen at the time of euthanasia.
  2. We have added the following sentence to the results section (page 7, line 218):

Assessment at the time of euthanasia demonstrated that the leukemia burden in the bone marrow was >50% (Supplemental Figure 1).

  1. We have added the following to the discussion section (page 8, line 290):

Another consideration is that we included both drug toxicity and leukemia-induced morbidity as events for our in vivo assessments. During determination of the MTD, the attribution to drug toxicity was clear with dactinomycin doses ≥75mg/kg, based on severe clinical symptomatology necessitating euthanasia. This differentiation was less evident when performing the EFS studies; however, given the high percentage of leukemia cells in the bone marrow at time of death during the EFS studies, we inferred that the primary cause of death in transplanted mice was related to leukemia. Serial assessment of blood parameters and detailed necropsy at the time of euthanasia may have assisted with further delineation.’

Reviewer 2 Report

Comments and Suggestions for Authors

In Figure 1 I still can not see the color precedent just a scale with percentage values, but without colors. So what red color or blue color means is not clear.

Author Response

Comment 1: In Figure 1 I still cannot see the color precedent just a scale with percentage values, but without colors. So what red color or blue color means is not clear.

Response 1: Thank you for noticing this omission. The format of the figure has been corrected to show the colour scale in PDF version.

Reviewer 3 Report

Comments and Suggestions for Authors

The authors have followed my suggestions. I do not have any further comment to the paper.

Author Response

Comment 1: The authors have followed my suggestions. I do not have any further comment to the paper.

Response 1: Thank you for your review and constructive feedback of our manuscript.

Reviewer 4 Report

Comments and Suggestions for Authors

I read the modified manuscript and  do not have further comments

Author Response

Comment 1: I read the modified manuscript and do not have further comments

Response 1: Thank you for your review and constructive feedback of our manuscript.